# Recent Advances on Membranes for Water Purification Based on Carbon Nanomaterials

**DOI:** 10.3390/membranes12100915

**Published:** 2022-09-22

**Authors:** Nikita S. Lazarenko, Valerii V. Golovakhin, Artem A. Shestakov, Nikita I. Lapekin, Alexander G. Bannov

**Affiliations:** Department of Chemistry and Chemical Engineering, Novosibirsk State Technical University, 630073 Novosibirsk, Russia

**Keywords:** carbon nanomaterials, membranes, carbon nanotubes, membrane purification, membrane technology, water, water purification, graphene, graphene oxide

## Abstract

Every year the problem of water purification becomes more relevant. This is due to the continuous increase in the level of pollution of natural water sources, an increase in the population, and sharp climatic changes. The growth in demand for affordable and clean water is not always comparable to the supply that exists in the water treatment market. In addition, the amount of water pollution increases with the increase in production capacity, the purification of which cannot be fully handled by conventional processes. However, the application of novel nanomaterials will enhance the characteristics of water treatment processes which are one of the most important technological problems. In this review, we considered the application of carbon nanomaterials in membrane water purification. Carbon nanofibers, carbon nanotubes, graphite, graphene oxide, and activated carbon were analyzed as promising materials for membranes. The problems associated with the application of carbon nanomaterials in membrane processes and ways to solve them were discussed. Their efficiency, properties, and characteristics as a modifier for membranes were analyzed. The potential directions, opportunities and challenges for application of various carbon nanomaterials were suggested.

## 1. Introduction

Membrane purification of water is one of the most intensive ways to improve the quality of water compared to other techniques used. In this review, we present the applications of membranes in both desalination processes and the treatment of wastewater from various types of pollutants related to organic pollutants, heavy metal salts, etc. Membrane processes for water purification and desalination include microfiltration (MF) [1], ultrafiltration (UF) [2], nanofiltration (NF) [3], membrane distillation (MD) [4,5,6], ion exchange membranes and direct osmosis (FO—forward osmosis; RO—reverse osmosis) [7,8] (Figure 1).

The separation principle in these processes is based on the different sizes of molecules and other objects (MF, UF and NF), the ionic charge of the molecules and membrane surface (NF, FO and RO), gradient of vapor pressure (MD) and electrical polarity (ion exchange membranes). MF, UF, NF and RO are well-established processes in which the separation takes place under hydraulic pressure. FO is a new process where the separation is carried out under osmotic pressure. The osmotic pressure is the hydrostatic pressure applied to the feed side of the membrane necessary to stop the flow of water through the membrane. In RO, water flows from the salt solution to the waterside of the membrane if a pressure higher than the osmotic pressure is applied to the salt side of the membrane.

MF is a pressure separation process that is widely used to concentrate, purify or separate macromolecules, colloids and suspended particles with an average molecular weight > 400 kDa from solution. MF membranes typically have a nominal pore size of the order of 0.1–1.0 µm and an operating pressure less than 2 bar, it is widely used in the food industry for applications such as wine, juice and beer purification, wastewater treatment and separation of plasma from blood for therapeutic and commercial applications.

UF is a low-pressure process, and the pore size of the membranes is in the range of 0.003–0.1 µm. It uses only the size exclusion principle to operate. Higher molecular weight species and suspended particles can be retained. Polymers that are both hydrophilic and hydrophobic can be used to create UF membranes. Even at 1–2 bar gage pressure, approximately 100% separation is attainable by UF since the solution’s osmotic pressure is not a significant concern. Due to the UF membrane’s small hole size, the heavy metal ions can easily travel through it. 

Another pressure-driven membrane separation method that is frequently used for water softening is nanofiltration (NF) (i.e., separation of divalent and monovalent cations). With pore sizes ranging from 1 to 10 nm, NF membranes have slightly larger pores than reverse osmosis membranes. The influence of the membrane charge should be taken into account in the transport theory because the membrane is frequently charged.

Now it is relevant to build mathematical models to develop the most effective membranes [9]. For example, in [10] considerable work has been performed to build the membrane model. Due to this, it is possible to create membranes of different types and shapes, taking into account a large number of factors, angles and layers, etc., leading to modeled and successfully implemented processes for CO_2_ water purification [11]. A model of the operation of the membrane contaminated under different conditions was also developed [12]. Models based on ion exchange in membranes are also being studied [13]. Based on the Donnan–steric-pore model, the influence of physicochemical characteristics was studied, it was shown that a large contribution is made by the support layer created on the basis of polymers [14].

The main problem in the development of membrane technologies is obtaining an inexpensive, stable, tunable, and multifunctional material for membrane fabrication. The most important problem with any kind of membrane is fouling (organic, bio, inorganic, reversible or irreversible, etc.). Better materials are needed for use as membranes. The capacity of such new materials is the key factor for industrial processes in order to reduce these types of fouling and lower their cost [15,16,17].

Carbon nanomaterials are of interest for implementation in membrane technologies due to their unique structural characteristics. This article reviewed current works on the introduction of carbon nanomaterials and related materials as membrane materials. This work is aimed at finding modern methods for the preparation and application of membranes. This paper may provide insight into the development of membranes based on carbon nanomaterials for water purification in the future.

## 2. Materials for Membrane Purification

The application of nanomaterials as modifiers for membranes in water purification is of interest due to its potential of scale-up and high efficiency. Nanotechnologies possess great potential to improve water and wastewater treatment, increasing treatment efficiency as well as increasing water supply through the safe use of unconventional water sources [18]. The advantages of nanomaterials are high specific area, fast dispersion ability, high reactivity, and sorption ability to various organic pollutants [19,20,21], heavy metals [22,23] and bacteria [24,25]. Carbon nanomaterials [26,27,28], nanomaterials based on metals and metal oxides [29,30,31], and composite nanoparticles with magnetic and ferromagnetic properties [32,33,34] are widely applied for water purification [35]. 

Carbon nanotubes, carbon nanofibers, fullerenes, graphene and graphite-like materials and their derivatives, as well as amorphous carbon composites, are related to the carbon nanomaterials (CNMs) [36,37,38] (Figure 2).

The main advantage of carbon nanomaterials is their porosity [40,41]. The surface of CNMs can be functionalized with various chemically active groups (hydroxyl and fluorinated alkyl groups [42], carboxyl [43], aldehyde [44], amino [45]) for impact on some specific pollutants [46,47]. Some carbon nanomaterials can be used as fillers for conventional membranes for removal of pollutants [48].

Nanomaterials based on metals and metal oxides are the class of materials consisting of one, two or three metals and/or their oxides [49,50]. Despite the wide variety of these materials, there are only few substances used for purification, such as zero-valent iron [51,52], titanium dioxide TiO_2_ [53,54], silver [55,56], and zinc oxide ZnO [55,57]. Mechanism of water purification includes adsorption, reduction, chemical decomposition, and chemical disinfection [35].

Nano- and micron-sized particles with magnetic properties are mainly represented by iron, cobalt, nickel, copper, and their oxides and alloys [58,59,60]. Magnetic materials are deposited on the shell, which can be made of inorganic particles (SiO_2_ and Al_2_O_3_) or organic particles (polymers or surfactants). This coating makes it possible to improve the chemical stability of particles, prevent the oxidation, and to provide the certain functions, such as selectivity of absorption of ions. One of the main advantage of such nanoparticles is their supermagnetism assisting the fast separation of pollutants from pure water under an external magnetic field, requiring less energy for achieving the set level of separation than non-magnetic particles [61].

Although it is supposed that the methods based on nanotechnologies are more expensive, in [35], the authors proposed cheaper and effective alternatives to traditional methods. Moreover, the methods based on the application of nanomaterials could be extremely important for meeting increasingly stringent water quality standards, especially for the removal of new pollutants and low levels of pollutions (low concentrations of pollutants) which are difficult to remove by traditional method. There are some ways to improve the efficiency of water purification using carbon nanomaterials. The materials were actively investigated due to the anti-biofouling properties of membranes that increase the operation period of membranes [62,63,64] (Figure 3). Biofilm formation and adhesion are reduced by modifying membranes with two-dimensional or zero-dimensional carbon-based nanomaterials or their modified substituents. Functionalization of nanofillers with different organic ligands or the composition of nanofiller with other materials, such as silver nanoparticles, enhances the bactericidal ability of composite membranes. Moreover, such membrane modifications reduce biofilm adhesion while increasing water permeability and salt/dye rejection.

## 3. Carbon Nanofibers

Membranes based on carbon nanofibers (CNFs) are already successfully used in water treatment technology [66]. CNFs are attractive for membrane water purification due to their relatively simple synthesis, chemical stability, and the low cost of CNFs with high specific area make them good candidates for the removal of pollutants mainly because of low diameter (below 100 nm) and high aspect ratio [67]. They possess the ability to remove toxic metals due to the multiple interactions that could be realized (for example, π–π interaction, electrostatic interaction, and surface complexation) [68]. In addition, they have a positive effect on the removal of organic substances. At the same time, CNF-based membranes can be operated in a wider pH range than conventional cellulose membranes [62]. 

Usually, in order to improve the dispersion of CNFs in the polymer matrix, the initial materials are subjected to modification. In [62], Tavakol et al. developed CNFs-based thin film composite membranes for direct osmosis technology. Carboxylation of CNFs was carried out to increase surface hydrophilicity and improve dispersion in the polysulfone matrix. At the same time, the use of functionalized CNFs positively affects the structure, morphology and performance of direct osmosis membranes (Figure 4).

The CNF loading improves the FO water flux of the fabricated membranes in both FO and PRO modes of operation. The FO water flux increases from 6.85 to 13.08 L·m^−2^·h^−1^ in FO mode and from 18.80 to 36.15 L·m^−2^·h^−1^ in PRO (active layer facing DS) mode by adding the CNFs from 0 to 0.3 wt.% to the membrane substrate. It was concluded that the membrane with the loading of 0.3 wt.% CNFs with the best FO performance with water flux of 13.08 L·m^−2^·h^−1^ in FO mode was the optimal FO membrane [62]

Another way to improve the properties of CNFs for desalination is by doping with heteroatoms. For example, in [69] the authors prepared a flexible N-doped CNF membrane modified with ZnO for the extraction of heavy metal ions (Pb^2+^, Cu^2+^, Cd^2+^). The use of this membrane in capacitive deionization (CDI) technology reduced Pb^2+^, Cu^2+^, Cd^2+^ by 99.42%, 68.46% and 70.36%, respectively. The results show that doped CNFs have great potential application for the removal of heavy metal ions. Moreover, the electrode material preparation method is simple, providing a cost-effective and feasible method for CDI for the removal of heavy metal ions.

The creation of composites also positively affects the water treatment properties due to their synergistic nature. In [70], a carbon nanofibers (CNFs) composite membrane with resorcinol-formaldehyde (RF) and activated carbon fiber (ACF) was synthesized for effective water removal from water–oil emulsion. The membrane had an asymmetric internal structure with a hydrophobic CNF-decorated microporous surface, which facilitated oil to pass and the rejection of water droplets from the emulsion. The dead-end separation test data indicated a flux of 426 ± 20 L·m^−2^·h^−1^ with the water removal efficiency of 99.7% and the permeate having water droplets with the size range of 37–78 nm when tested against a water–oil emulsion of 10% (*v*/*v*) comprising 700–1700 nm sized water droplets.

## 4. Activated Carbon

Activated carbon is currently one of the most common membrane materials used in water purification. Although it is not a nanomaterial, its role in the enhancement of membrane process is significant. Activated carbon belongs to a wide range of carbonized materials with a high degree of porosity and large surface area. This property make it possible to find many applications in the environment and industry for removal, extraction, separation of various compounds in the liquid and gas phases [71]. One of its most important advantages over other carbon-based materials is its cheapness and simplicity of production.

### 4.1. Adsorption Mechanism of Activated Carbon

The graphite-like structure of activated carbon makes it an excellent material for the adsorption of organic compounds, especially those containing an aromatic functional group. Activated carbon also has a high adsorption capacity for small organic compounds such as DBPs and some pharmaceutical compounds [72]. Adsorption usually occurs through π–π interactions between the adsorbate and the surface of the activated carbon, but can also occur through hydrogen bonding and Van der Waals force. This material is also capable of cation/anion–π interactions with metal species, but this process is generally considered weak, especially for unmodified activated carbon. For example, Figure 5 shows the mechanism of metal ion adsorption on the surface of activated carbon.

The adsorption capacity of activated carbon can be influenced by factors such as: pH of the medium, ion concentration, temperature and concentration of other types of pollutants, which can have both positive and negative effects depending on the adsorbate [74]. The physical size of compounds interacting with activated carbon also affects adsorption capacity and efficiency, with larger compounds being able to cause pore clogging and limit the diffusion of smaller particles [75].

### 4.2. Improving the Properties of Activated Carbon as a Membrane Component by Its Modification

In addition to the activation method, it is possible to control the properties of the resulting material by modifying it by depositing metal nanoparticles or metal oxide inside the pores of the activated carbon. Metal-impregnated activated carbons are usually obtained by the reduction in metal salt solutions or by the direct adsorption of pre-prepared nanoparticles [76]. These composites have shown a high adsorption capacity to contaminants including heavy metals as well as halides. Thus, a comparison of unmodified and silver nanoparticle-coated activated carbon in the bromine and iodine removal process showed that the silver-impregnated material was significantly better at removing bromides (95% compared to 26% for the unmodified sample) [77]. This material also showed increased removal efficiency of total dissolved organic carbon when used in combination with an improved coagulation process (77% compared to 67%).

Iron nanoparticles in a number of experiments on the removal of As(III), As(V), Hg(II) and Pb(II) salts from solution for unmodified sample were 0.73, 0.09, 2.61 and 3.06 mg/g, respectively, while adsorption on iron-impregnated activated carbon was an order of magnitude higher and was 4.67, 4.50, 4.57 and 4.35 mg/g, respectively [78]. Activated carbons impregnated with silver nanoparticles have also demonstrated an outstanding bactericidal effect, making them extremely promising materials for contaminated water treatment [79]. However, despite the high-performance characteristics, the use of activated carbon with coated metal nanoparticles is simply not possible, due to the gradual washout of nanoparticles from the membrane surface. 

It was also shown in [80] that microwave radiation can be used to modify activated carbon. Although microwave heating is not a direct chemical modification, it can result in physical and chemical changes in activated carbon associated with an increase in internal surface area, a reduction in the average pore size by creating new, smaller pores and a change in surface chemistry [81]. The use of microwave heating may be preferable due to two main advantages: lower energy consumption and the duration of the modification process.

In addition, a relatively new way of modifying activated carbon can be called biological modification. Biologically activated carbon filters are formed by introducing certain bacteria that are trapped in the pore channels of the activated carbon. Then, if optimal conditions and organic nutrients are available, the bacteria can multiply, repopulating the entire inner surface of the carbon. Subsequently, when the membrane is exposed to water containing organic substances and, in particular, complex macromolecular organic compounds, adsorption on the surface of the activated carbon occurs, and further, under the influence of bacteria, the adsorbed matter decomposes into carbon dioxide, water and simpler organic compounds. These decomposition products can then be easily adsorbed onto the carbon surface and removed from the water. Usually, ozone or UV treatment is applied to the dissolved contaminants before filtration with biologically activated carbon [82]. Additional treatment resulted in a 40% and 30% improvement in water purification from trihalomethanes (THMs) and haloacetic acids (HAs), respectively, observed when ozone or UV treatment was applied prior to treatment with bioactivated carbon [83]. This process creates highly reactive groups in organic compounds, increasing their bioavailability and ease of biodegradation by bacteria. Granular activated carbon inoculated with different strains of heterotrophic nitrifying bacteria showed improved ammonia removal efficiency, an increase of 2.8–4 times compared to conventional activated carbon [84]. A pilot study using similar membranes in a wastewater treatment plant showed their effectiveness in removing THM and HA, with removal efficiencies of 45% and 80% for each, respectively [85]. However, the effectiveness of such materials is severely limited by the pH of the medium, and the biological activation of the carbon leads to an accumulation of biomass inside the material, which must be periodically removed to maintain optimum efficiency. Heat treatment and various chemical reactions can be used to regenerate biologically activated charcoal, but these can lead to loss of structure of the biologically activated charcoal and also require the use of hazardous, environmentally polluting reagents. As a viable alternative, low-frequency ultrasound has been introduced as a new method for regeneration of biologically activated carbon [86]. With this approach, the ability to remove dyes and ammonium from aqueous solutions was recovered.

### 4.3. Activated Carbon Modified Membranes

The addition of activated carbon as an adsorbent to water treatment membranes can have a beneficial effect on the efficiency of water treatment. Thus, the addition of granular activated carbon to various gravity membranes has been shown to reduce the antiepileptic drug carbamazepine and the human metamizole biotransformation product formaminoantipyrine by at least 88% and 92%, respectively. The improved removal of formaminoantipyrine, despite lower adsorbability compared to carbamazepine, can be attributed to the biodegradation process. It has been shown that organic micropollutants can be effectively removed by ultraviolet irradiation. The proposed membrane system effectively traps bacteria and particles at 5.1 and 1.6 log removal value (LRV), respectively [87].

Ahmad et al. [88] fabricated a highly efficient hybrid symmetric membrane made of activated carbon deposited on ceramics for oily wastewater treatment (Figure 6). The hybrid membrane had complex micro-nano-channel networks, achieving twice the porosity of the Al_2_O_3_ membrane. The oil removal efficiency of the hybrid membrane reached 99.02%.

Li et al. [89] describe a support-free membrane made of activated carbon, binder, porogen, and conductive agent, the reagents were subjected to directional carbonization under pressure. The selection of the reagent composition and carbonization conditions allows the selection of the required adsorption, surface and membrane characteristics, while creating a surface to deposit the catalyst on the carbon membrane substrate in combination with the maintenance of electrochemical technology. This technology has made it possible to achieve a versatile purification of liquids contaminated primarily with organic substances. 

Bae et al. [90] developed an activated carbon membrane modified with carbon filaments for wastewater and potable water treatment. The carbon filaments have a positive effect on the service life of the membrane by preventing sedimentation and accumulation of particles (Figure 7). The adsorption capacity of this membrane in the phenol removal process was comparable to that of commercially available activated carbon, but with significantly better stability.

Yu et al. [91], in a pilot plant, showed that membranes pre-coated with powdered activated carbon can remove different types of organic compounds with different molecular weights and hydrophobicity. In addition, this membrane allowed the improvement of the performance of the UF system, not only increasing the possible initial flow volume but also improving the recovery rate of backwashing.

Abuabdou et al. [92] were able to obtain flat sheet membranes made of polyvinylidene fluoride with the addition of activated carbon powder. The resulting material showed significantly higher performance compared to unmodified polyvinylidene fluoride. The addition of activated carbon powder effectively increased the rate of contaminant removal and had a positive effect on membrane fouling. Increasing powdered activated carbon to a certain value had a positive effect on the efficiency of chemical oxygen removal, color removal and NH_3_-N removal. The composition of (14.9 wt.%) polyvinylidene fluoride and (1.0 wt.%) powdered activated carbon was shown to be optimum and to increase the cleaning efficiency to 36.63% chemical oxygen removal, 49.5% color removal and 23.84% NH_3_-N.

## 5. MXene

MXenes are the family of 2D transition metal carbides, carbonitrides and nitrides. This class of materials can be considered as related material to graphene. The directional synthesis of MXenes enables their use in various fields, e.g., membranes, reinforced composites, bio- and gas sensors, etc. [93,94]. Two-dimensional materials such as graphene oxide, transition metal dichalcogenide and MXene have recently started to be used for membrane fabrication. They are characterized by high solvent permeability and accurate molecular selectivity [95,96,97] (Figure 8).

Li et al. reported on an ultra-thin GO membrane with good hydrogen separation selectivity [98]. Nanoporous 2D graphene membranes have also been used for desalination and nanofiltration [99,100].

Ding et al. [95] fabricated a 2D lamellar membrane from MXene. The 2D MXene membranes were prepared using synthesized MXene nanosheets filtered on porous AAO (anodic aluminum oxide). The water permeability of MXene membranes is approximately 5 and 10 times higher than membranes not filtered on AAO. In [94] it was shown that desalination of salt water by membrane distillation is promising due to 100% theoretical salt removal capability.

Zhang et al. [101] created a hybrid membrane based on MXene graphene oxide and a commercially available PVDF membrane, the evaporation of saline water reached 92.5% without salt accumulation for 10 h in 10 wt.% NaCl solution. Zhang et al. [94] developed empirical modeling based on experimental results:J(T,S) = 8.82 × 10^−3^T^2^ − 42.49 × 10^−2^T + 18.21 × 10^−3^TS + 27.13 × 10^−2^S + 6.64(1)
where T and S represent the feed temperature (°C) and solar illumination density (sun), respectively.

Wang et al. [102] modeled the contribution of permeability to total transport through direct and bypass nanotunnels:(2)$$f_{str}=\frac{P_{str}}{P_{str}+P_{tor}}=\frac{Zd_{p}}{Zd_{p}+3l_{s}\;\exp\left(-\frac{E_{a,g}}{RT}\right)}$$
(3)$$f_{tor}=\frac{P_{tor}}{P_{str}+P_{tor}}=\frac{3l_{s}\;\exp\left(-\frac{E_{a,g}}{RT}\right)}{Zd_{p}+3l_{s}\;\exp\left(-\frac{E_{a,g}}{RT}\right)}$$
where *f_str_* and *f_tor_* are fractional gas permeance through direct and circuitous nanotunnels, respectively (Figure 9).

## 6. Carbon Nanotubes

Additional processing is required for graphite materials as the carbon layer is able to peel from the substrate under harsh conditions (harsh solvents, high pressures, etc.). Results are known for the modification of substrates with carbon nanotubes (CNTs), polydopamine (PDA), polyethyleneimine (PEI), plasma treatment, etc. Possible membranes using CNT-based composites include CNT functionalized ZIF-8-impregnated poly(vinylidene fluoride-co-hexafluoropropylene) [103], and CNT@MOF5-incorporated poly(vinylidene fluoride-hexafluoropropylene) [104], etc.

As the multi-walled carbon nanotubes (MWCNT) structure is represented in the form of entangled agglomerates, there is a great difficulty in the interaction with the polymer matrix. Due to the weak interfacial interaction, the problem of preparing a homogeneous dispersion of MWCNT arises [105]. The solution to this problem is the pre-modification of MWCNT, followed by the formation of carboxylic and hydroxyl groups on the surface [106]. Fictionalization occurs as a result of chemical reactions such as esterification, amination, silanization, etc. [107,108]. 

In [109], Du et al. obtained the membrane from hollow tubes based on CNTs on electrospun PAN fibers (Figure 10), which has a significant advantages on its permeability compared to commercial PAN membranes (permeability was 7.3 times higher).

Xue et al. [106] obtained three types of modified MWCNT with carboxyl, hydroxyl, and amino groups, which were incorporated in the separating layers of NF membranes. The amino group showed higher stability than the carboxylic group due to different adhesion to the PA matrix. MWCNT modified with hydroxyl groups had the highest PWF of 41.44 L·m^−2^·h^−1^·bar^−1^ and Na_2_SO_4_ rejection of 97.6% (Figure 11).

Wang et al. designed and fabricated a bilayer material (MWCNT/SiO_2_) for high efficiency water evaporation under solar energy. The MWCNT acted as a photothermal material with high absorption capacity in sunlight, and the SiO_2_ substrate was required to form the desired surface microstructures of the MWCNT, acting as a thermal barrier [110].

Double-walled carbon nanotubes (DWCNT) is a more promising membrane material than single-walled carbon nanotubes (SWCNTs) because DWCNT is 18.9% more water permeable than the latter [111] (Figure 12).

Tijing et al. fabricated superhydrophobic nanofiber membranes containing MWCNT. The presence of MWCNT resulted in an increase in membrane permeability. In addition, compared to commercially available PVDF membranes, the membranes obtained had higher porosity (485%) and increased strength (121% and 360% depending on the MWCNT mass content). The permeability of these membranes reached 24–29.5 L·m^−2^·h^−1^, compared with 18–18.5 L·m^−2^· h^−1^ of commercial membranes [112] (Figure 13).

## 7. Graphene and Graphene Oxide

Graphene is one of the most popular carbon materials for research in the 21st century [113]. The potential of graphene (and graphene related materials) as a key material for advanced membranes stems from two main possible advantages of this atomically thin 2D material: permeability and selectivity. Graphene-based membranes are also hypothetically attractive in terms of concentration and contamination polarization, as well as the chemical and physical stability of graphene. However, further research needs achieving of these theoretical advantages. In addition, improvements in design and manufacturing processes to create efficient and cost-effective graphene-based desalination devices are a matter of discussion. Finally, membranes are only one part of desalination systems, and current processes are not optimized to take full advantage of the higher selectivity and permeability of graphene [9,114].

Active research is being carried out in the following areas: creation of graphene-based membranes and their application as stimulus-responsive separating materials [115]; investigations of the effect of graphene functionalization on the physical and chemical properties of the membranes [116]; creation of nanofiltration membranes with salt removal and fouling protection [117]; membranes based on graphite carbon nitride with high catalytic activity [118]; development of reduced graphene oxide membranes with low ion permeability [119]; creating the membranes based on the polymer composites with the inclusions of both conventional graphene and its oxide [120,121,122]; membranes made of graphene nanosheets and nanoribbons [123]; microscopic settings of graphene oxide framework for membrane separation [124]; pervaporative membranes from GO [125]; membranes which are able to work in the cross-flow mode; and the influence of the flow together with the solvent on the membrane [28].

A method to create a highly permeable selective nanofiltration membrane from thin-film composites (TFC) based on graphene quantum dots (GQDs) [126,127] was also developed. The main idea is to modify the GQDs with amino/sulfone groups (GQDs_N/S) as building blocks. Hydrophilic sulfo groups are introduced to give the membrane a stronger internal polarity. They break the densely packed structure of GQDs to form an expanded interparticle space with increased free volume for faster water transport inside the membrane. The separation properties with high water permeability (9.8 L·m^−2^·h^−1^·bar^−1^) and a rejection rate of 97% relative to sodium sulfate, suggests that water permeability was increased by more than 2 times compared to conventional TFC without reducing membrane selectivity.

Graphene was obtained from soybean oil by the CVD method [128], which was wet applied to a commercial polytetrafluoroethylene (PTFE) membrane. The CVD process in air allows the growing of continuous graphene films with a high density of nanocrystalline grain boundaries on a polycrystalline Ni substrate, which are desirable as channels permeable for water vapor. The obtained graphene was superimposed in several layers with overlapping grain boundaries. Such overlaying allows for the rejection of various metal salts, surfactants and oils. The obtained sample was tested in the following way: the membrane with the size of 4 cm^2^ was processed in seawater for 72 h. The capacity was 0.5 L·day^−1^.

In [129], the authors investigated the effect of chemical treatment on graphene. The treatment was carried out in the reactor milisicundum gasification under the action of O_2_/O_3_ and high temperature. This etching method makes it possible to create subangstrom size pores. Membranes with the addition of such graphene have good CO_2_/N_2_ and CO_2_/CH_4_ separation.

Membranes based on graphene-polyurethane composites for desulfurization of water and gas were created in [130]. Two-dimensional “egg yolk-shell” structures of graphene were embedded into the polyurethane matrix obtaining a mixed matrix membrane. With a graphene content of 0.8%, an optimum enrichment factor of 4.32 with a permeability flux of 1411 g·m^2^·h was achieved. Membranes for the separation of water–oil mixtures were also created based on their “superhydrophobicity” and “superoleophilicity” (SHSO) [131]. It was noted that when adding graphene oxide to such membranes, separation reaches 99%.

Scientists have also developed a Janus membrane evaporator made of porous graphene fibers and cellulose fibers by light filtration [132]. Such membranes are capable of collecting 95.6% due to multiple reflections and converting them into thermal energy, with a maximum water evaporation rate of 1.40 kg·m^−2^·h^−1^ with a corresponding efficiency of 88%. In addition, this membrane is capable of purifying seawater from metal ions and leaching ions from man-made wastewater.

There are membranes of large surface area (84 cm^2^) based on multilayer defect graphene doped with B,N-groups on a substrate of α-Al_2_O_3_ (pore size 100 nm). A continuous nanometer (50 nm) film of chitosan containing adsorbed (NH_4_)_3_BO_3_ was deposited on the aluminum oxide. Subsequent pyrolysis in the presence of hydrogen transforms chitosan into multilayer defective B,N-doped graphene. Partial removal of the doping B and N atoms with H_2_ during pyrolysis leads to the formation of subnanometric pores due to atomic vacancies. Above 95% removal efficiency of NaCl and KCl from brackish water was achieved at a permeate flow of 24.3 L·m^−2^·h^−1^ at 10 bar [133,134]. 

The study of nanopore geometry has determined that the effective area of nanopores plays a critical role, and to better understand the influence of this parameter, the aspect ratio and equal diameter of noncircular pores are calculated based on different methods such as equal area, equal perimeter and hydraulic diameter for each case. In the case of non-circular nanopores, the results show that the aspect ratio and hydraulic diameter as well as the equal area method can adequately reflect the relevant trends, while the equal perimeter method is not able to correctly predict them [135].

GO-based membrane was synthesized by the authors in [136] by the application of a Fe^3+^ or Al^3+^ trivalent metal cation to cross-link the GO nanosheets overlaid on the PVDF membrane support layer. The hydrophilicity of the cation-modified GO increased with increasing amount of cation introduced. Prior to fabrication, the PVDF membrane was pre-conditioned by filtration with 200 mL of deionised water and then the GO was applied to the membrane surface by pressure filtration to obtain a clean membrane. To obtain a stable and suitable membrane, Al^3+^ or Fe^3+^ (in concentrations of 0.001 M, 0.01 M or 0.1 M) was added as a crosslinking agent to 100 mL of deionised water with 1 mg or 2 mg GO (corresponding to GO loading 220 mg/m^2^ or 440 mg/m^2^). Finally, the residual unbound Al^3+^ or Fe^3+^ was removed from the synthesized GO membrane by filtration with deionized water for ten minutes. It was found that the Fe^3+^ cross-linked GO membrane exhibited a higher flux and removal efficiency of natural organic compounds compared to the Al^3+^ cross-linked GO membrane (Figure 14).

The purification characteristics of GO-based membranes vary widely. This is due to the presence of oxygen-containing functional groups (carboxyl (-COOH), hydroxyl (-OH) and epoxy (-COC-)). Research carried out in [137] showed that COOH-GO with a large number of negative charges showed excellent antifouling properties for trypan blue (dye) due to the electrostatic interaction effect. However, such membranes have worse water transport properties than membranes with hydroxyl (-OH) and hydrogen (-H) groups. These properties are discussed by the authors in [138], explaining the large steric-geometric structure of the COOH-GO-based membranes shows resistance to the head of water flow. At the same time, the OH group promotes water transport by “pulling” water molecules through the nanosheet layer. This property is due to the relatively strong interaction of the OH group with water. The hydrogen atom also promotes water transfer, mainly due to its steric structure with low resistance. The results show that when designing GO membranes with high water flux it will be strategic to remove the edge functional groups of COOH while retaining a mixture of edge functional groups of OH and H (Figure 15).

In [139], the authors created a membrane that makes it possible to carry out the electrical control of water penetration. The membrane consists of graphene oxide on a porous silver substrate with a thin layer of gold (10 nm). The metal played the role of electrodes. The thin layer of gold is sufficiently porous and has little or no effect on the properties of ultrafast water permeability. The Ag-GO-Au membrane was made according to following technique. The graphene oxide suspension was obtained by exfoliation of graphite oxide powder in water using ultrasonic treatment. Then the suspension was applied on a porous metallic «Sterlitech» silver membrane (pore size 0.2 µm; diameter 13 mm) by vacuum filtration. The Ag-GO membrane was bonded with epoxy resin to a polyethylene terephthalate tape, which is required to prevent shorting between the electrodes when gold is applied. 

In [140,141], the problem with the delamination and swelling of graphene oxide in aqueous medium was solved by adding β-cyclodextrin (β-CD)-polyacrylic acid (PAA). The bonding of graphene nanosheets with β-CD is due to covalent bonds. This modification increases the interlayer spacing *d*, at the same time creating chiral spacer and adsorption centers between the GO layers. This adds the property of rejecting dyes with increasing filter size. The PAA penetrates into the cavities of the CD-GO composite, forms a strong “brick-mineral” configuration due to hydrogen bonds, fills defects in the structures, and increases the negative charge improving molecular selectivity. A number of experiments with dye-contaminated water found that the rejection and permeability of the membrane depended on the amount of PAA added. Thus, for GSP, the flow through the membrane decreased from 21 L·m^−2^·h^−1^ (96.7% rejection) and for GSP-5 to 13.4 L·m^−2^·h^−1^ (99% rejection).

The authors of [142,143,144] used the idea of functionalization of graphene oxide by β-CD to make polyamide thin-film nanocomposite membranes. Such membranes have good characteristics for reverse osmosis, antifouling and antibacterial properties. These membranes were prepared by the reaction of amide bonding with ethylenediamine grafted β-CD and GO. The obtained nanosheets were incorporated into thin-film composite membranes by interfacial polymerization of m-phenylenediamine in an aqueous solution containing β-CD-f-GO nanosheets with trimesoyl chloride. The water permeability flux and antibacterial activity of the membranes improved by 38% and 46%, respectively, compared to the unmodified membrane. The total fouling coefficient decreased by 26% and the flow coefficient increased by 83% compared to the unmodified membrane. In addition, the membrane has a long-term stability of 144 h for reverse osmosis and resistance to Cl_2_.

Nitrogen groups can create strong electrostatic interactions with metal ions and lead to a selectivity of FGOMs (functionalized graphene oxide membranes) for mono/divalent cations up to 90% and 28.3% in single and binary solution, which is more than 10 times higher than the selectivity of graphene oxide membranes (GOMs). First-principles calculations confirm that the ion selectivity of FGOMs is due to the difference in binding energies between metal ions and polarized nitrogen atoms. In addition, ultra-thin FGOMs with a thickness of 50 nm can have a high water flux of up to 120 mol·m^−2^·h^−1^ without compromising the rejection of nearly 99.0% NaCl solution [145].

In [146], the authors printed the membrane on an inkjet printer using a suspension of graphene oxide. They obtained a large-area membrane (>100 cm^2^), which was printed on commercial polysulfone (PS). These membranes were used for the selective separation of gases (H_2_, He, CO_2_, N_2_). By introducing CO_2_-philic agents, the separation efficiency of CO_2_/N_2_ (up to 70). H_2_/CO_2_ separation selectivity was up to 25. Such membranes are capable of stable operation for a long time.

The authors of [147] considered the idea of introducing membranes in various branches of modern power engineering. They suggest introducing composite membranes in biofuel production processes, since the selection ability of membranes will allow rejecting by-products. 

It was also found that proton selectivity increases with channel sizes smaller than 8 A. Such dimensions are close to the end of the radius range of hydration of I- and II-valent cations. The Grottus selectivity mechanism provides a proton-switching membrane for fuel cells and the possibility of more stable liquid hydrogen storage [148].

In [149], C. Cheng et al. evaluated the change in the size of nanochannels in graphite oxide membrane layers by exposing it to iodic acid vapor with different exposure durations. The change in nanochannel size allows for better ion rejection. Thus, the rejection rate increased from 29% to 57%. At the same time, this method caused a membrane permeability defect due to a decrease in surface hydrophilicity and narrowing of the nanochannel. In addition, the researchers note that it is necessary to determine the optimal time of exposure to iodic acid on the membrane to accurately control the size of the nanowell at the sub-nanometer scale and simultaneously prevent the reduction in water permeability when adjusting the ion selectivity. The authors of [150] considered the possibility of obtaining environmentally friendly graphene oxide from gum arabic (GGA) and the creation of membranes based on polyvinyl sulfone (PPSU) (Figure 16).

The effect on the physicochemical properties of membranes based on PPSU by adding small amounts (0–0.25 wt.%) of GGA was studied. The results showed that a membrane prepared using 0.15 wt.% GGA could provide maximum permeability characteristics (119 ± 3 L·m^−2^·h^−1^) and retention potential (88%), whereas optimal relative flow (11%) was obtained using the lowest amount of GGA (0.05 wt.%). In addition, although the modified membranes showed a comparable hydrophilic character (contact angle within ~50°), their performance was slightly different. This indicates that the permeability characterization of any membrane is a complex issue and is not always related to the value of the contact angle. Other surface properties such as pore size, porosity, charge, and roughness can interact and determine overall membrane performance.

In [151], polypyrrole nanoparticles were intercalated to graphene oxide membranes (GOM) to increase stability and decrease membrane swelling (Figure 17). Polypyrrole adsorbs well the particles of various dyes, resulting in an improvement in rejection properties from 60% to 97%. It also increases nanochannels and improves water permeability by up to 14 times compared to GOM without polypyrrole.

In addition, polypyrrole improves the mechanical stability of the membrane due to the rigid hydrogen and N–H bonds. Studies with almost identical results have been conducted by [152]. The authors of [153] created a membrane with a rejection rate of up to 1000 dye molecules at 295 L·m^−2^·h^−1^·bar^−1^.

In [154], the authors learned how to align circular pathways (which are longer than the membrane thickness) inside the graphene oxide membrane by introducing doped zirconium. It allows creating the necessary Z-patterns, which improves then through it and maintains high selectivity. The Zr-GO composite allows the membrane to self-assemble with uniform and controlled nanochannel lengths.

Magnesium oxide nanoparticles (MgO NPs) were combined with a modified membrane of carboxymethyl cellulose (CMC) macrostructure and graphene oxide nanosheet (GO) to form a hybrid composite as a modified surface for anti-fouling membranes and supercapacitors. Comparing the available composites, the GO-CMC-MgO composite has the best performance [155].

When water passes through the membrane, some of the water evaporates due to friction. In [156], the authors created a nanocapillary membrane with nanopores and nanochannels. Several such membranes can charge a 1-F supercapacitor and power LEDs (2 V·20 mA).

High-performance graphene oxide membrane created by introducing positively charged polyethylenimine grafted GO (GO-PEI) into negatively charged GO nanosheets [157]. It was found that the additional GO-PEI component changes the surface charge, improves hydrophilicity and expands the nanochannels. The water permeability of such membranes reaches 89 L·m^−2^·h^−1^·bar^−1^. Because of the electrostatic interaction, it can selectively remove positively charged methylene blue at pH 12 and negatively charged methyl orange at pH 2, with a removal rate of over 96%. The high and cyclic water permeability and highly selective removal of organics can be explained by the synergistic effect of the controlled nanochannel size and electrostatic interaction in response to the ambient pH.

A new 2D bimetallic sulfide/N-doped reduced graphene oxide composite (FeCoS-N-rGO) was developed by self-assembling MOFs and GO nanosheets followed by carbonization and further sulfation, then a 2D/2D catalytic membrane (FeCoS-NGC) was fabricated by intercalating FeCoS-N-rGO into adjacent graphite carbon nitride (g-CN) [158]. In such a composite membrane, the permeation flux increased from 76 L·m^−2^·h^−1^ to 632 L·m^−2^·h^−1^. In addition, FeCoS-NGC was used to efficiently remove other contaminants, including tetracycline, bisphenol A, rhodamine B, and perfluorooctanoic acid. Functional groups play an important role in catalytic water purification (Figure 18).

In [159], the effect of GO on the repair of defects in the g-C_3_N_4_ structure was investigated. This resulted in a continuous g-C_3_N_4_-GO membrane that exhibited excellent H_2_ permeability (H_2_ permeability: 2.16 × 10^−7^ mol·m^−2^·s^−1^·Pa^−1^, H_2_/CO_2_ selectivity: 39) and good stability (Figure 19).

Membrane based on graphene oxide with theanine (reducing agent) and tannic acid (crosslinking agent) is able to maintain a stable state longer (not delamination) and has a water permeability of more than 10,000 L·m^−2^·h^−1^·bar^−1^. It had an s rejection rate of up to 100% rhodamine B and methylene blue, and was sufficiently chemically resistant to acids and bases for several months [160].

Graphene oxide can also be used as a cell for the catalyst, thus creating a strong composite membrane [161]. Such a membrane has both separating and catalytic properties. Such a membrane is used in the persulfate-based Fenton reaction, but the persulfate itself is not capable of removing tough, harmful organic pollutants. The resulting composite membrane has excellent stability for continuous contaminant removal with a high efficiency above 90% and a stable flow of about 40 L·m^−2^·h^−1^ in the sequential filtration mode for 24 h.

The overview of the articles devoted to various types of carbon nanomaterials used in membranes is shown in Table 1.

## 8. Challenges and Opportunities

There is a lot of research aimed at improving the performance of membranes by modifying them with various carbon materials. Some nanocarbons, such as AC nanoparticles, have long been known to science and have been used in commercial membrane filters. Nowadays, this work is focused on finding and optimizing new raw material sources and procedures for the production of the material. At the same time, modern materials such as MXene are still at the beginning of intensive research, but the possibility of using this material as a membrane material has already been investigated in a number of papers. The completely wide range of carbon nanomaterials can be applied to improve membrane technology and, as in the case of AC, the transition of some of the most promising materials from the scientific to the industrial environment is only a matter of time. 

Further direction of investigation of CNTs for membrane technology will be directed to application of doped and gradient-porous nanotubes. The use of heat-treated and ball-milled CNTs will be also promising to be studied extensively. At the same time application of special non-conventional types of CNT modification, e.g., plasma functionalization will bring an enhancement of oxygen content in the material that is also attractive for CNT-based membranes.

Graphene oxide application in membranes will be devoted to directed creation of certain functional groups with the enhancement of wettability with water and interaction with pollutants. However, the problem will rise at the formation of porous reduced graphene oxide and the methods keeping the pore structure in a combination with oxygen-containing functional groups will make it possible to improve the GO-based membranes. 

## 9. Conclusions

In conclusion, it can be said that the reviewed works sufficiently reveal the potential for composite membranes with carbon nanomaterials. The permeability and rejection exceed the values of commercial membranes, making it possible to conclude that the composite membranes with the inclusion of carbon materials are promising. Possibly, the opportunity to modify the membranes with materials nanomaterials will allow them to be used, preserving all the advantages of membrane processes and at the same time achieving high purification of wastewater. The main advantage for the use of carbon nanomaterials is the possibility of their targeted synthesis. The selection of techniques for the production of carbon material allows the achievement of high purification efficiency for a specific pollutant nature. The possibility of improving the obtained carbon material by heteroatom doping, chemical and physical modification is important. Here we review the modern approach to modify the membranes using various carbon nanomaterials. Five main groups of carbon nanomaterials were presented in terms of their applicability for enhancement of membrane efficiency. 

The reviewed works fully reveal the potential of composite membranes with both graphene and graphene oxide. In order to obtain some specific properties, it is possible to add different compounds to improve the membrane properties at the synthesis stage of graphene materials. It is possible that the use of such technologies in the future will provide a huge advantage to graphene membranes against the commercial membranes because of the ease of creation of these membranes, high chemical resistance in aggressive environments and high desalination capacity. At the same time, CNTs used in membrane technologies need to be functionalized. There is a difference between behavior of SWCNTs, DWCNTs, and MWCNTs in membranes. MWCNTs induced the increase in hydraulic resistance of membranes compared to DWCNTs and SWCNTs. Membranes based on CNFs can potentially be used, however there is a lack in experimental data. The simplicity of synthesis of CNFs and their cheaper price can be the key factors in their application. At this moment, since these nanomaterials are produced in small amounts in laboratories, the more advanced technologies should be used for synthesis and modification of novel carbon nanomaterials in industry, including their directed control of texture properties and surface chemistry. The real use of the materials considered in the review paper is the purification of water from heavy metals, drugs, dyes, oil and other dangerous pollutants.

Modern approaches and the desire to reuse wastewater, as well as society’s high demand for clean water, are the main drivers for research into the most efficient and affordable ways of water treatment and purification. Membrane filters modified with carbon materials are already far superior to simple polymeric membranes, and their ability to flexibly adapt to pollutants makes them potentially one of the most relevant technologies for treating water from pollutants of any nature. This paper will be useful for provide extended information on the application of novel carbon nanomaterials in water purification.

## Figures and Tables

**Figure 1 membranes-12-00915-f001:**
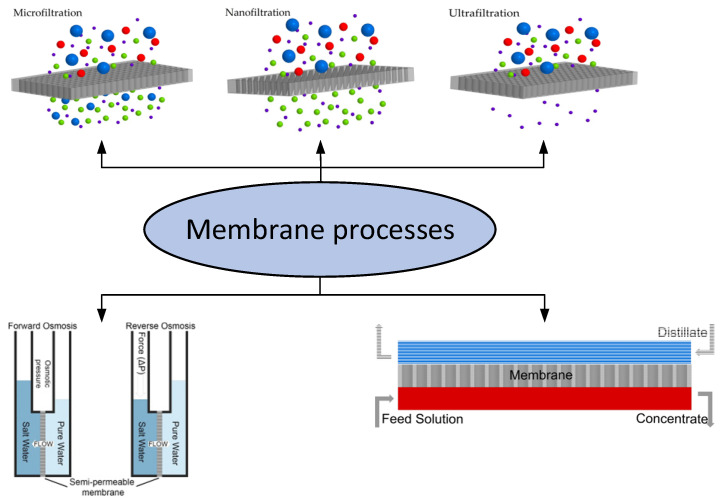
Types of membrane processes.

**Figure 2 membranes-12-00915-f002:**
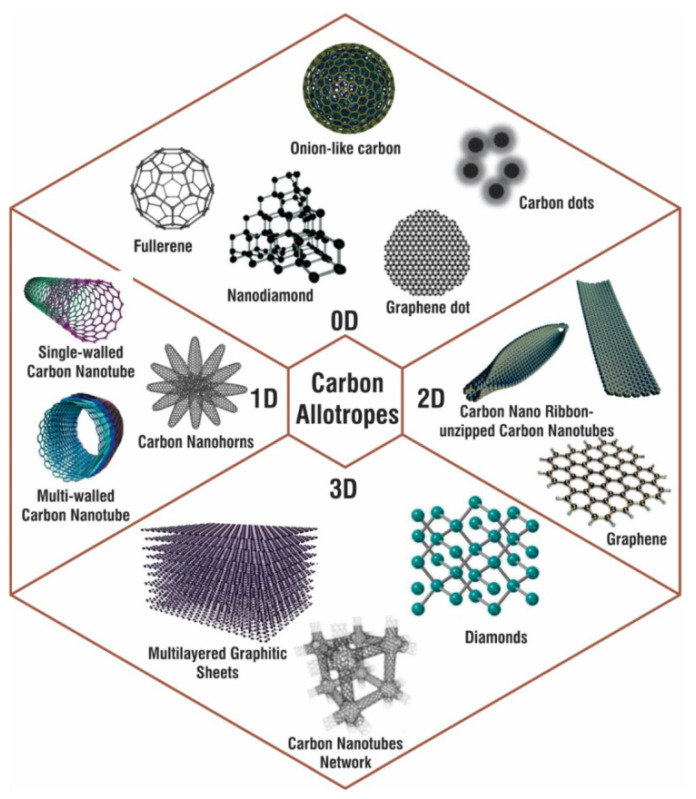
Various carbon allotropes with examples for 0D, 1D, 2D, and 3D carbon nanostructures. Diagram taken from [39].

**Figure 3 membranes-12-00915-f003:**
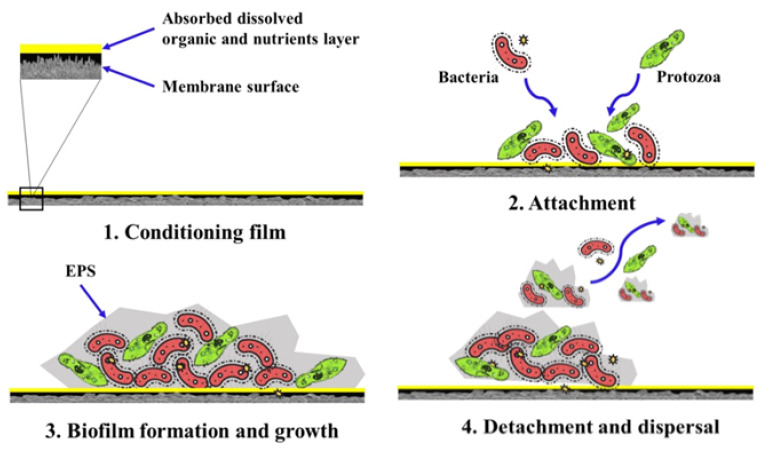
The formation and development of biofouling (**1**→**4**) in a membrane-based process for water and wastewater purification. Diagram taken from [65].

**Figure 4 membranes-12-00915-f004:**
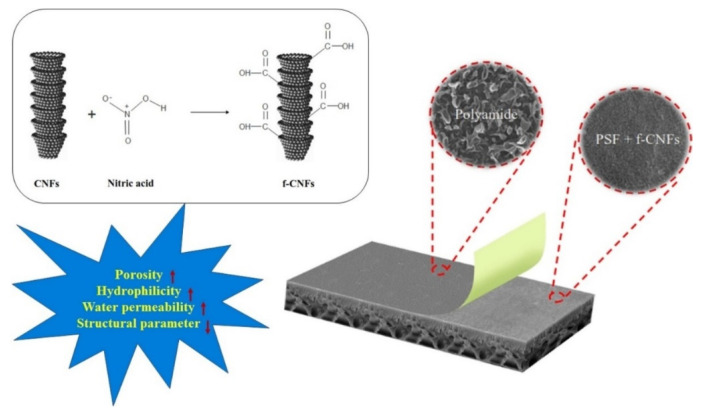
Thin film composite membrane for forward osmosis made of polysulfone matrix/functionalized CNFs with the polyamide rejection layer. Diagram taken from [62].

**Figure 5 membranes-12-00915-f005:**
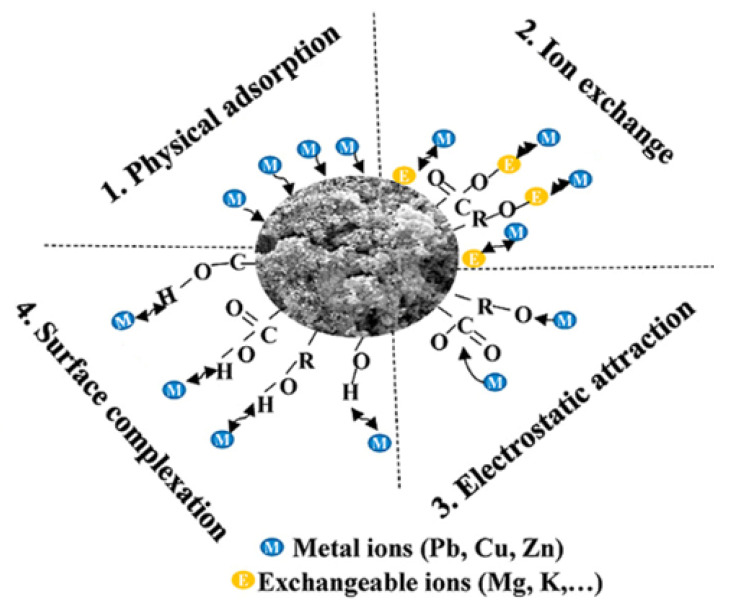
Schematic mechanism of metal ion adsorption by AC surface. Diagram adapted from [73].

**Figure 6 membranes-12-00915-f006:**
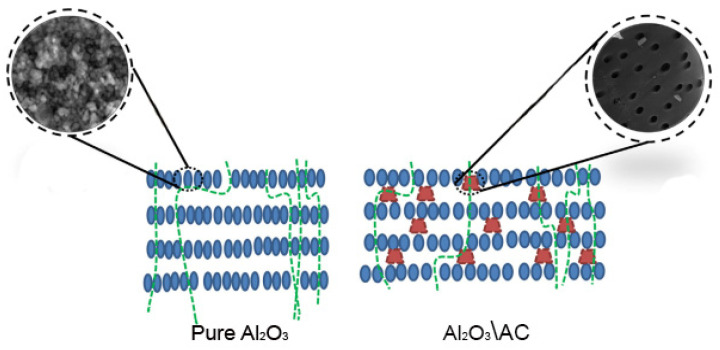
Comparison of Al_2_O_3_ and Al_2_O_3_/activated carbon hybrid membrane. The inclusion of carbon in the aluminum oxide matrix can increase the porosity of the structure, thereby causing additional microchannels for water to pass through the membrane. This leads to an increase filtration and adsorption efficiency. Diagram taken from [88].

**Figure 7 membranes-12-00915-f007:**
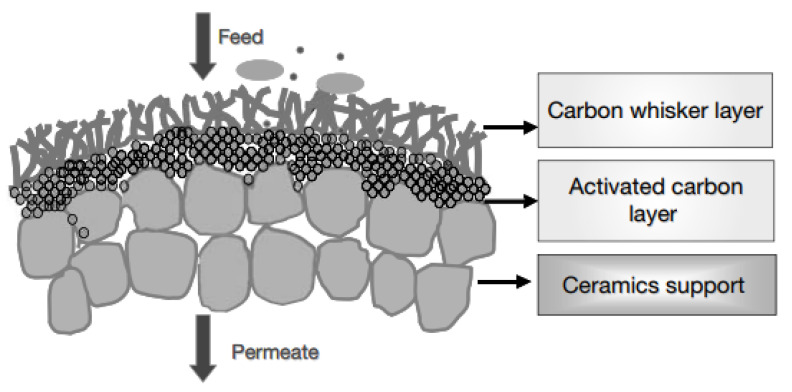
Scheme of a carbon fiber-modified activated carbon membrane. Diagram taken from [90].

**Figure 8 membranes-12-00915-f008:**
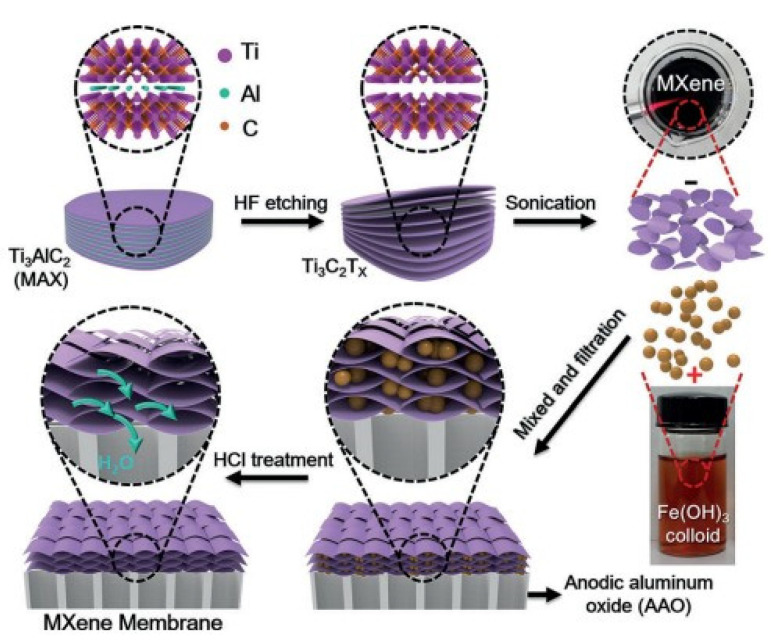
MXene membrane preparation, scheme taken from [95].

**Figure 9 membranes-12-00915-f009:**
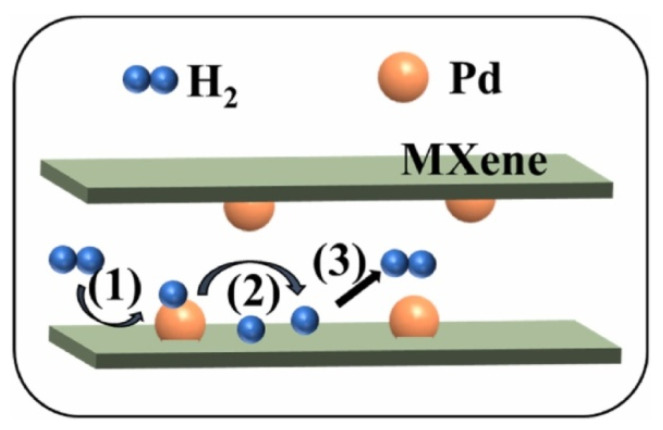
Schematic illustration of the H-spillover process, diagram taken from [102].

**Figure 10 membranes-12-00915-f010:**
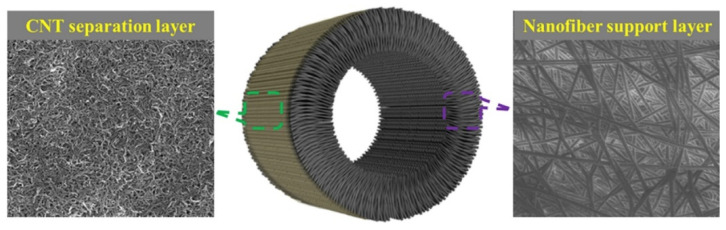
Images of conductive CNT/nanofiber composite hollow fiber membrane with electrospun support layer. Diagram taken from [109].

**Figure 11 membranes-12-00915-f011:**
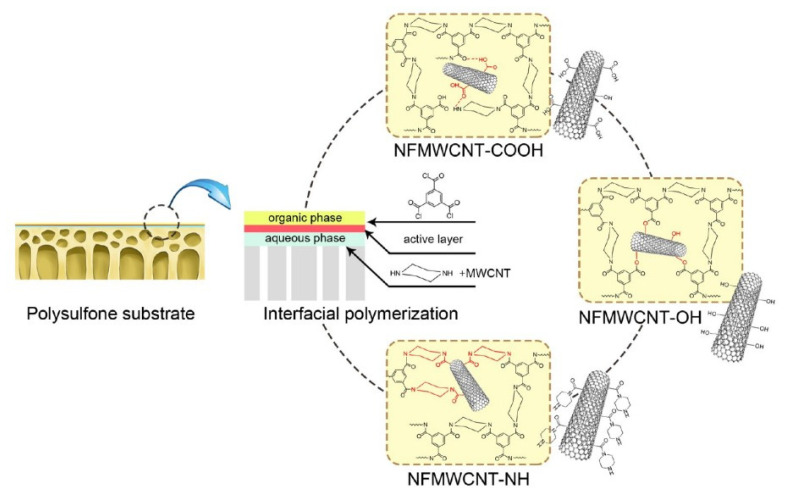
Schematic illustration for the preparation of different nanocomposite NF membranes via interfacial polymerization [106]. Reprinted with permission from { Xue, S.M.; Xu, Z.L.; Tang, Y.J.; Ji, C.H. Polypiperazine-Amide Nanofiltration Membrane Modified by Different Functionalized Multiwalled Carbon Nanotubes (MWCNTs). ACS Appl. Mater. Interfaces **2016**, 8, 19135–19144, doi:10.1021/acsami.6b05545}. Copyright 2022 American Chemical Society.

**Figure 12 membranes-12-00915-f012:**
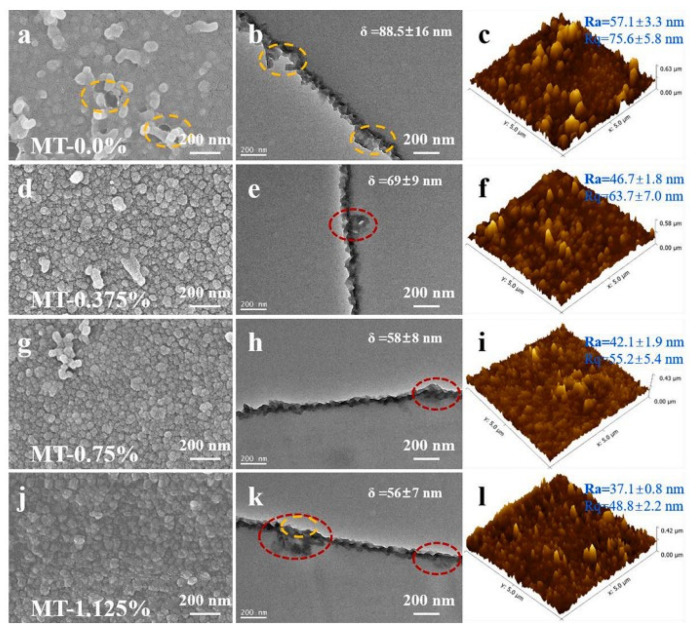
SEM images of the top view of nanofiltration membranes (NFMs) (left), TEM image of cross-sectional views for PA layers (middle) and AFM topographical images of NFMs (right): (**a**–**c**) MT-0.0%, (**d**–**f**) MT-0.375%, (**g**–**i**) MT-0.75%, and (**j**–**l**) MT-1.125%. Yellow circles indicate defects and red circles represent HF-MoS_2_, diagram taken from [97].

**Figure 13 membranes-12-00915-f013:**
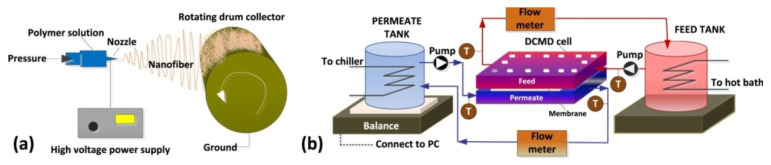
Schematic illustrations of the (**a**) electrospinning set-up and (**b**) direct contact membrane distillation system, diagram taken from [112].

**Figure 14 membranes-12-00915-f014:**
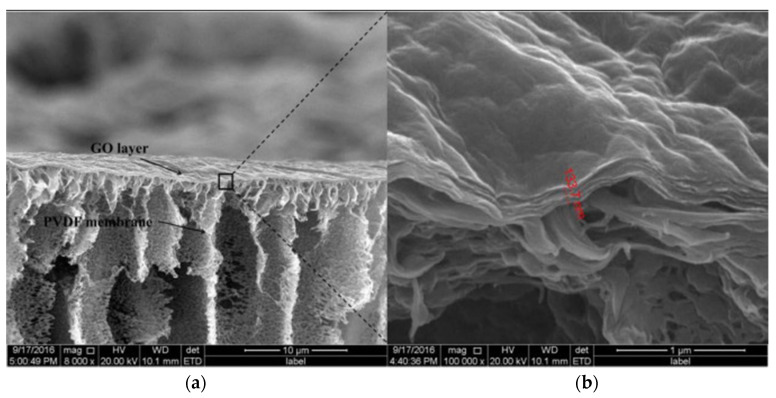
**=** SEM images of the GO membrane (**a**,**b**) [136].

**Figure 15 membranes-12-00915-f015:**
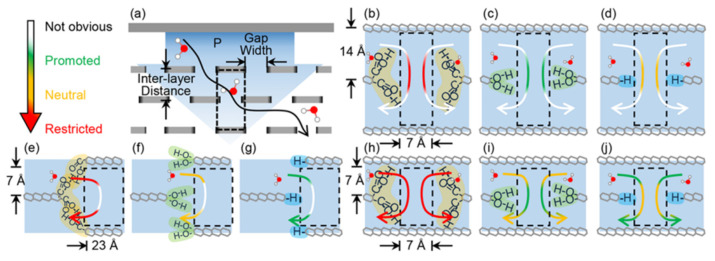
Summary of the individual influences of the three kinds of edge functional groups on water transport under different geometric membrane structures. (**a**) Illustration of the membrane system. Geometric structures of the membranes are (**b**–**d**) 7 × 14, (**e**–**g**) 23 × 7, and (**h**–**j**) 7 × 7, and their edge functional groups are (**b**,**e**,**h**) –COOH, (**c**,**f**,**i**) –OH, and (**d**,**g**,**j**) –H. Green, orange, red, and white parts of the arrows illustrate the promoted, neutral, restricted, and not obvious influences from the edge functional groups, respectively [138]. Reprinted with permission from {Qiu, R.; Yuan, S.; Xiao, J.; Chen, X.D.; Selomulya, C.; Zhang, X.; Woo, M.W. Effects of Edge Functional Groups on Water Transport in Graphene Oxide Membranes. ACS Appl. Mater. Interfaces **2019**, 11, 8483–8491, doi:10.1021/acsami.9b00492}. Copyright 2022 American Chemical Society.

**Figure 16 membranes-12-00915-f016:**
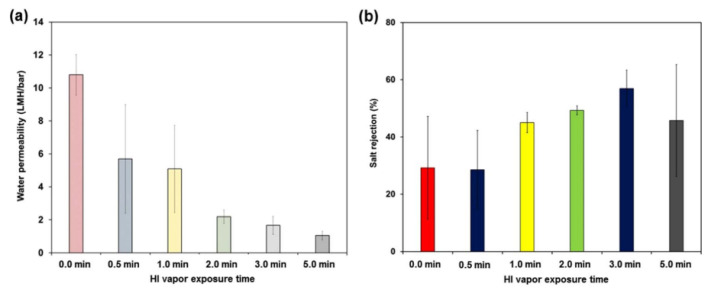
(**a**) Water and (**b**) salt rejection of GO membranes, diagram taken from [150].

**Figure 17 membranes-12-00915-f017:**
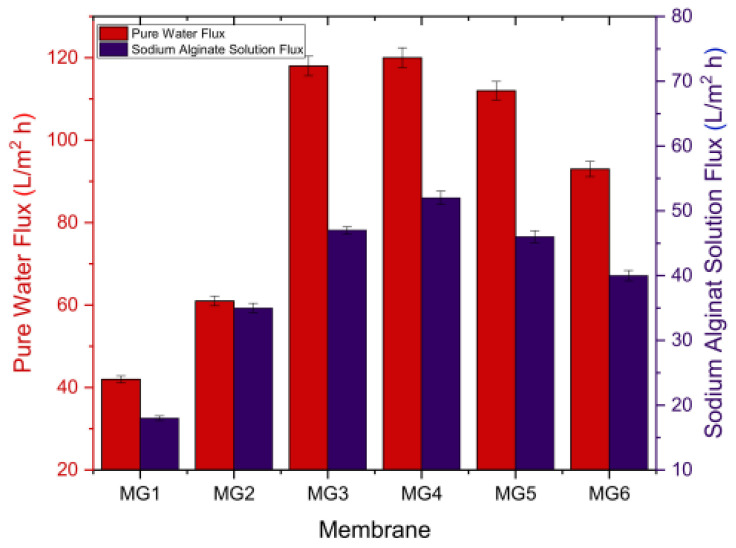
Pure water and NaAlg solute alginate solution flux of synthesized membranes, diagram taken from [151].

**Figure 18 membranes-12-00915-f018:**
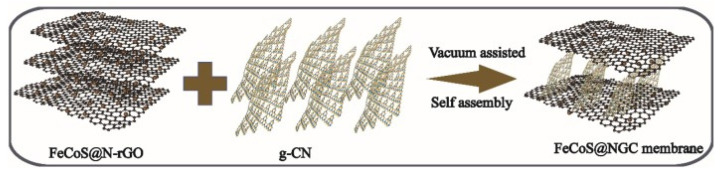
The self-assembly process of FeCoS-NGC membrane, scheme taken from [158].

**Figure 19 membranes-12-00915-f019:**
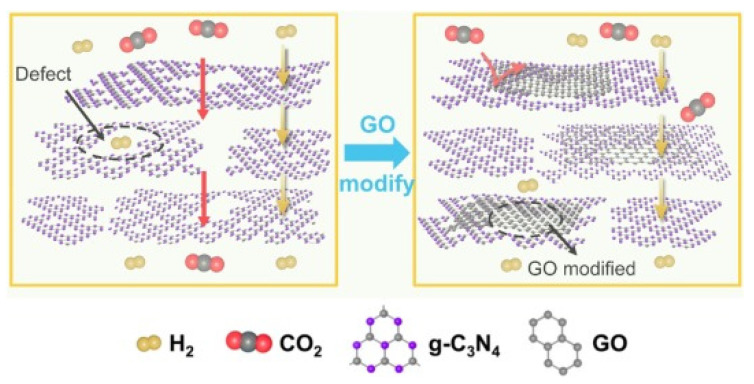
Illustration of gas molecular transport through the GO-modified g-C_3_N_4_ nanosheets membranes, scheme taken from [159].

**Table 1 membranes-12-00915-t001:** Application of carbon nanomaterials and related materials in membranes.

Membrane Material	Carbon Material	Pollutant	Water Flow	Rejection	Pressure	Ref.
Polyvinylidene fluoride	Granular activated carbon	Carbamazepine	4.3–5.1 L·m^−2^·h^−1^	88%	32.5 mbar	[87]
Polyvinylidene fluoride	Granular activated carbon	Formylaminoantipyrine	4.3–5.1 L·m^−2^·h^−1^	92%	32.5 mbar	[87]
Al_2_O_3_	Powdered activated carbon	Oil	18 L·m^−2^·h^−1^	99%	5 bar	[88]
Polyvinylidene fluoride	Activated carbon powder	NH3-N	69.9 L·m^−2^·h^−1^	20%	0.6 bar	[92]
Polyvinylidene fluoride	Activated carbon powder	Colour	89.3 L·m^−2^·h^−1^	51.5%	0.6 bar	[92]
Polyvinylidene fluoride	Activated carbon powder	Oxygen	32.2 L·m^−2^·h^−1^	51.5%	0.6 bar	[92]
Al_2_O_3_	MXene	Bovine serum albumin	>1,000 L·m^−2^·h^−1^·bar^−1^	~100%	0.1–0.6 MPa	[95]
Cellulose ester	SWCNTs	Oil/water	100,000 L·m^−2^·h^−1^·bar^−1^	99.95	0.01 MPa	[162]
Polysulfone	DWCNTs	3.4-dihydroxyphenylalanine	0.35–37.44 mmol·m^−2^·h^−1^	98–99%	5 bar	[163]
Polyethersulfone	MWCNTs	NaCl, MgSO_4_, Na_2_SO_4_	6–14 kg/m^2^·h	20–80%	4 bar	[164]
Polysulfone matrix	Carboxylated carbon nanofibers	NaCl	13.08 L·m^−2^·h^−1^	94.5%	N/A	[62]
Resorcinol-formaldehyde -activated carbon fiber matrix	CuO-CNF	Oil	426 L·m^−2^·h^−1^	99.7%	5	[70]
Porous carbon nanofiber matrix	ZnO modified N-doped porous carbon nanofibers	Pb^2+^	N/A	99.42%	N/A	[69]
Porous carbon nanofiber matrix	ZnO modified N-doped porous carbon nanofibers	Cu^2+^	N/A	68.46%	N/A	[69]
Porous carbon nanofiber matrix	ZnO modified N-doped porous carbon nanofibers	Cd^2+^	N/A	70.36%	N/A	[69]
Thin film composites (TFC)	Graphene quantum dots	Na_2_SO_4_	9.8 L·m^−2^·h^−1^·bar^−1^	97%	1 bar	[126]
Polytetrafluoroethylene (PTFE)	Graphene	Various metal salts, surfactants and oils	0.5 L·day^−1^	N/A	N/A	[128]
Polyurethane	Graphene	Desulfurization of water and gas	1,411 g·m^2^·h	99%	N/A	[130]
Cellulose fibers	Graphene	Metal ions and leaching ions from man-made wastewater	1.40 kg·m^−2^·h^−1^	95.6%	N/A	[132]
α-Al_2_O_3_	B,N-doped graphene	NaCl, KCl	24.3 L·m^−2^·h^−1^	95%	10	[133]
β-cyclodextrin, polyacrylic acid	Graphene oxide	Metal ions	21 L·m^−2^·h^−1^	96.7%	N/A	[140]
Polyvinyl sulfone	Graphene oxide	Various metal salts	119 ± 3 L·m^−2^·h^−1^	88%	N/A	[151]
Polypyrrole	Graphene oxide	Various metal salts	295 L·m^−2^·h^−1^·bar^−1^	N/A	N/A	[152]
Polyethylenimine	Graphene oxide	Organic compounds	89 L·m^−2^·h^−1^·bar^−1^	96%	-	[157]
Bimetallic sulfide (FeCoS)	N-doped graphene oxide	Tetracycline, bisphenol A, rhodamine B, and perfluorooctanoic acid	632 L·m^−2^·h^−1^	N/A	N/A	[158]
g-C_3_N_4_	Graphene oxide	H_2_/CO_2_	2.16×10^−7^ mol·m^−2^·s^−1^·Pa^−1^	N/A	N/A	[159]
Theanine	Graphene oxide	Rhodamine B and methylene blue	10,000 L·m^−2^·h^−1^·bar^−1^	100%	N/A	[160]

## Data Availability

Not applicable.

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
