# Peer review of "Recent Advances on Membranes for Water Purification Based on Carbon Nanomaterials"

_membranes, 2022, doi:10.3390/membranes12100915_

Round 1

Reviewer 1 Report

This review presents the applications of carbon nanomaterials to the membrane water purifications. The topic is a promising approach for modern membrane technologies, and this review will provide the potential readers with valuable information. Therefore, I think that this review should be published in this journal. However, I have to raise some points that might be reconsidered in a revised version.

1. In the manuscript, the description “Figure X” should be noted in brackets.
Ex.) Page 4, line 115: … membranes [57-59] Figure 3.

2. Is correct the caption of Figure 5?

3. In Section 6, it is difficult for me to understand the first paragraph.

4. The descriptions of author names of the cited papers are not unified in the main text.
Ex. Li et al. (page 11, line 420), L. Ding et al. (page 11, line 423), J. Zhang et al. (page 11, line 429), B. Zhang (page 11, line 431), Qinsheng Wang et al. (page 12, 438), etc.

5. I cannot find the explanations for some abbreviated words.
Ex. UNTs (page 5, line 158), AC (page 7, line 275), GA (page 12, line 455), MWNT (page 13, line 465), DWCNT (page 13, line 493), SWCNT (page 13, line 493), NFMs (page 14, line 497), MT (page 14, line 498), GGA (page 19, line 702)

6. Page 20, lines 756-759: This paragraph provides the description about the gas separation. Why is this description required in this review?

7. The present form of conclusions are not in accordance with the main text. The present conclusions mainly provide the summary for graphene. I would like to recommend more general conclusion.

8. I found some typos. Please recheck the original manuscript carefully.
Page 3, line 81: grapheme
Page 3, line 88: hydroxyl and and specific
Page 4, line 132: and low cost CNFs
Page 5, line 176: makes it possible it to
Page 14, line 517: 21st centu

Author Response

Reviewer 1.

This review presents the applications of carbon nanomaterials to the membrane water purifications. The topic is a promising approach for modern membrane technologies, and this review will provide the potential readers with valuable information. Therefore, I think that this review should be published in this journal. However, I have to raise some points that might be reconsidered in a revised version.

  1. In the manuscript, the description “Figure X” should be noted in brackets.

Ex.) Page 4, line 115: … membranes [57-59] Figure 3.

Thank you for the comment! The descriptions “Figure X” were noted in brackets.

  1. Is correct the caption of Figure 5?

The caption of Figure 5 was changed, the link has been changed to the correct one.

  1. In Section 6, it is difficult for me to understand the first paragraph.

Section 6 was changed.

  1. The descriptions of author names of the cited papers are not unified in the main text. Ex. Li et al. (page 11, line 420), L. Ding et al. (page 11, line 423), J. Zhang et al. (page 11, line 429), B. Zhang (page 11, line 431), Qinsheng Wang et al. (page 12, 438), etc

All information except surname was deleted.

  1. I cannot find the explanations for some abbreviated words. Ex. UNTs (page 5, line 158), AC (page 7, line 275), GA (page 12, line 455), MWNT (page 13, line 465), DWCNT (page 13, line 493), SWCNT (page 13, line 493), NFMs (page 14, line 497), MT (page 14, line 498), GGA (page 19, line 702)

AC was changed to a full version of this term.  UNT was delited, DWCNT MWCNT, SWCNT were added, Added all transcripts (page 19, line 702)

  1. Page 20, lines 756-759: This paragraph provides the description about the gas separation. Why is this description required in this review?

We noted the high degree of purification of water using a membrane based on nitride-doped graphene, but also it was decided to show that the use of such materials has a positive effect on the purification of gases with the same high degree of purification.

  1. The present form of conclusions are not in accordance with the main text. The present conclusions mainly provide the summary for graphene. I would like to recommend more general conclusion.

The Conclusions were changed to more general ones.

  1. I found some typos. Please recheck the original manuscript carefully.

Page 3, line 81: grapheme

Thank you for remark. It has been changed.

Page 3, line 88: hydroxyl and and specific

Thank you for remark. It has been changed.

Page 4, line 132: and low cost CNFs

Page 5, line 176: makes it possible it to

Thank you for remark. It has been changed.

Page 14, line 517: 21st centu

Thank you for remark. It has been changed.

Reviewer 2 Report

Authors aim to review the most recent advances on membranes based on carbon nanomaterials for applications in water purification. The idea could be interesting, and the revision is correctly updated, nevertheless it focusses too much on graphene and graphene oxides, revising too less other materials. Most of the abstract is purely introduction. It must be rewritten to explain what authors aimed to review in their work. Finally, conclusions section is very poor and centres only in some of the carbon nanomaterials reviewed.

Some other comments follow:

-        Line 26: … the most intense ways

-        Line 33: MD is governed by differences in the vapour pressure, not really hydrophobicity

-        Line 40-50 are too much generalist and too basic level, well know concepts

-        Figure 1:  several mistakes in the texts on the figure. On the other side, ion exchange scheme represented is only one of the possible uses of ion exchange membranes, perhaps authors refer to electrodialysis, a more specific term for a membrane process based on the use of ion exchange membranes.

-        Line 63: fouling is a very strong problem in membrane application as hard as membrane material.

-        Line 72: I wonder you mean application of nanomaterials in the fabrication of membranes aimed for water purification. The nanomaterials you are reviewing never are the only compound of membranes, normally only additives in the membrane formulation

-        Line 100: shell?

-        Line 110: redundant quality

-        Check for typos and grammar mistakes all across the document (112, 132, 134,…)

-        Figure 3: this figure only makes sense if comparing the mechanism of anti-biofouling of CNM with usual membranes. On the other side, it is supposed the steps should be orderly placed (1 to 4)

-        Lines 148-153: results here commented are a particular case for a particular condition, it must be conveniently referenced.

-        Section 4 could be eliminated as does not fit the scope of the review. If so, only those cases where activated carbon is used to modify or improve membranes

-        Lines 420 and 422: why naming author of referenced works? better use only surname. In any case, use same decision through the paper

-        Line 453: I´m not sure what does PEI in the carbon nanotubes section, in any case, all interesting properties of PEI come from amine groups

-        Line 493: explain abbreviations first time used (DWCNT)

-        Lines 511-515: graphene is not a CNT

-        Line 779: simpler is not the key factor for their use of carbon nanomaterials.

Author Response

Reviewer 2.

Authors aim to review the most recent advances on membranes based on carbon nanomaterials for applications in water purification. The idea could be interesting, and the revision is correctly updated, nevertheless it focusses too much on graphene and graphene oxides, revising too less other materials. Most of the abstract is purely introduction. It must be rewritten to explain what authors aimed to review in their work. Finally, conclusions section is very poor and centres only in some of the carbon nanomaterials reviewed.

Some other comments follow:

  1. Line 26: … the most intense ways

Thank you for comment. It was changed.

  1. Line 33: MD is governed by differences in the vapour pressure, not really hydrophobicity

Thank you for comment! However, in the original reference (Boretti A. et al. Outlook for graphene-based desalination membranes // npj Clean Water. Nature Research, 2018. Vol. 1, № 1.) it was mentioned that the principle of separation in MD is based on hydrophobicity.

  1. Line 40-50 are too much generalist and too basic level, well know concepts

This part of section was rewritten.

  1. Figure 1:  several mistakes in the texts on the figure. On the other side, ion exchange scheme represented is only one of the possible uses of ion exchange membranes, perhaps authors refer to electrodialysis, a more specific term for a membrane process based on the use of ion exchange membranes.

Figure was corrected according to the remark.

  1. Line 63: fouling is a very strong problem in membrane application as hard as membrane material.

Thank you for remark. This is a significant problem which is not the main aim of this work. However, we have decided to mention this briefly in the Introduction.

  1. Line 72: I wonder you mean application of nanomaterials in the fabrication of membranes aimed for water purification. The nanomaterials you are reviewing never are the only compound of membranes, normally only additives in the membrane formulation.

It was rewritten. The application is for modification of membranes.

  1. Line 100: shell?

Yes, it is shell. Taken from original articleВзято с оригинала https://www.sciencedirect.com/science/article/pii/S1385894715015181

  1.  Line 110: redundant quality

It was changed.

  1. Check for typos and grammar mistakes all across the document (112, 132, 134,…)

Thank you for remark. It had been checked.

  1. Figure 3: this figure only makes sense if comparing the mechanism of anti-biofouling of CNM with usual membranes. On the other side, it is supposed the steps should be orderly placed (1 to 4)

Figure 3 was changed. The section devoted to biofouling was changed.

  1. Lines 148-153: results here commented are a particular case for a particular condition, it must be conveniently referenced.

The appropriated reference was added.

  1. Section 4 could be eliminated as does not fit the scope of the review. If so, only those cases where activated carbon is used to modify or improve membranes

The section 4 has been changed and formatted to be more relevant to the topic.

  1. Lines 420 and 422: why naming author of referenced works? better use only surname. In any case, use same decision through the paper

All information except surname was deleted. All names were formatted the same way throughout the paper.

  1.  Line 453: I´m not sure what does PEI in the carbon nanotubes section, in any case, all interesting properties of PEI come from amine groups

This paragraph was deleted from the article.

  1. Line 493: explain abbreviations first time used (DWCNT)

It has been added.

  1. Lines 511-515: graphene is not a CNT

Thank you for remark. The paragraph was deleted.

  1. Line 779: simpler is not the key factor for their use of carbon nanomaterials.

Thank you for remark. It has been corrected.

Reviewer 3 Report

Lazarenko et al. have reviewed recent advances on membranes for water purification based on carbon nanomaterials. I recommend acceptance of manuscript following revision as per following points:

Minor comments

Line 36-37: “applied to 36 the salt side” please replace salt side with feed side as salt side makes FO process as specific for salt only.

The authors discussed MF, UF, and NF in brief but not MD and ED; it would be interesting for the reader if the MD discussions were added. While discussing MD following works can be used: 10.1016/j.scitotenv.2021.150692, 10.1016/j.jece.2021.105818,

Major comments

What is new in this review compared to already available reviews in literature.

Have the authors only considered desalination and not wastewater? This should be made clear in introduction

If Figure 1 is adapted, provide a reference of the source. Moreover, in the Figure, the nanofiltration membrane pore size is more than the ultrafiltration membrane as per permeate side representation. 

The authors should also include a section (may be brief) on composites made from CNT and used to make membranes. While discussing this following works can be used: 10.1016/j.jece.2021.106560, 10.1016/j.colsurfa.2022.128918

“The membranes for water purification based on carbon nanomaterials are presented” A separate section includes the membranes economics based on carbon nanomaterials (suggestion).

A table showing the different contaminant can be tackled with best possible carbon nanomaterials reported can be represented and their limitations as well.

Please include a section on opportunities and challenges in the area of review.

Grammar should be reviewed. The linguistic flow might use some improvement.

Throughout the manuscript, certain errors must be changed.

Author Response

Reviewer 3.

Lazarenko et al. have reviewed recent advances on membranes for water purification based on carbon nanomaterials. I recommend acceptance of manuscript following revision as per following points:.

  1. Line 36-37: “applied to 36 the salt side” please replace salt side with feed side as salt side makes FO process as specific for salt only.

The order of words was changed.

  1. The authors discussed MF, UF, and NF in brief but not MD and ED; it would be interesting for the reader if the MD discussions were added. While discussing MD following works can be used: 10.1016/j.scitotenv.2021.150692, 10.1016/j.jece.2021.105818,

The references were added.

  1. What is new in this review compared to already available reviews in literature.

This review provides an overview of a range of carbon nanomaterials, without concentrating on a single type of material used to modify membranes. The aim of the authors was to give the most general insight into the potential applications of carbon materials in the fabrication of membranes

  1. Have the authors only considered desalination and not wastewater? This should be made clear in introduction

Information was added in introduction, you can find topics about both of this process during the text.

  1. If Figure 1 is adapted, provide a reference of the source. Moreover, in the Figure, the nanofiltration membrane pore size is more than the ultrafiltration membrane as per permeate side representation.

Figure 1 is not adapted and it is original. The effect of representation you mentioned is due to the viewing angle (the plane is more inclined), with the particle size corresponding to the type of filtration.

  1. The authors should also include a section (may be brief) on composites made from CNT and used to make membranes. While discussing this following works can be used: 10.1016/j.jece.2021.106560, 10.1016/j.colsurfa.2022.128918

The references were added.

  1. “The membranes for water purification based on carbon nanomaterials are presented” A separate section includes the membranes economics based on carbon nanomaterials (suggestion).

Unfortunately, there are almost no data on membranes economics based on carbon nanomaterials. Therefore, this section cannot be presented in the paper.

  1. A table showing the different contaminant can be tackled with best possible carbon nanomaterials reported can be represented and their limitations as well.

The table was added.

  1. Please include a section on opportunities and challenges in the area of review.

Grammar should be reviewed. The linguistic flow might use some improvement.

Thank you for remark. The section on opportunities and challenges was added. English was checked.

  1. Throughout the manuscript, certain errors must be changed.

The checking was carried out. The changes have been made.

Round 2

Reviewer 2 Report

Abstract remains unchanged, so my previous comment stands: “Most of the abstract is purely introduction. It must be rewritten to explain what authors aimed to review in their work”.

-        The driving force for MD is the gradient in vapour pressure at both membrane sides. Certainly membrane must be hydrophobic to get a functional process, but hydrophobicity is not the cause of the separation

-        Figure 1: ion exchange is not a membrane process but a type of membrane material.

-        Fouling (bio, organic, inorganic, reversible or irreversible, or all together) is always a major problem whatever membrane material or process, certainly better materials are needed to get more efficient process but at the end the capacity of such new materials to reduce fouling at economically acceptable cost is the key factor to industrial process success.

-        Ref. 98: Li et al. in text is correct

-        Conclusions: carbon nanomaterials or nanocarbon materials?

-        Still poor conclusions

Manuscript needs to be improved

Author Response

Authors thank reviewer for valuable comments.

The comments are presented below.

Abstract remains unchanged, so my previous comment stands: “Most of the abstract is purely introduction. It must be rewritten to explain what authors aimed to review in their work”.

Thank you for remark.  The abstract was changed.

-        The driving force for MD is the gradient in vapour pressure at both membrane sides. Certainly membrane must be hydrophobic to get a functional process, but hydrophobicity is not the cause of the separation

The “hydrophobicity” was deleted. The driving force of vapor pressure was added.

-        Figure 1: ion exchange is not a membrane process but a type of membrane material.

The part of Figure 1 in the part of exchange membrane was deleted. New Figure 1 was added.

-        Fouling (bio, organic, inorganic, reversible or irreversible, or all together) is always a major problem whatever membrane material or process, certainly better materials are needed to get more efficient process but at the end the capacity of such new materials to reduce fouling at economically acceptable cost is the key factor to industrial process success.

Thank you for valuable remark. The description of problem of fouling was added. “The important problem of any kind of membrane is fouling (organic, bio, inorganic, reversible or irreversible, etc.). The better materials are needed to be used for membranes. The capacity of such new materials is the key factor for industrial process in order to reduce these types of fouling and lower their cost.”

-        Ref. 98: Li et al. in text is correct

Thank you for remark.  It was changed.

-        Conclusions: carbon nanomaterials or nanocarbon materials?

Thank you for remark. Of course it is carbon nanomaterials.

-        Still poor conclusions

Thank you for remark. The conclusions were rewritten.

Manuscript needs to be improved

Reviewer 3 Report

The authors have addressed all the comments. The manuscript may be accepted for publication. 

Author Response

Authors would thank reviewer for valuable comments. All the comments were taken into account.

Round 3

Reviewer 2 Report

Manuscript has been reasonably improved